# Successful Eltrombopag Therapy in a Child with MYH9-Related Inherited Thrombocytopenia

**DOI:** 10.3390/children9121839

**Published:** 2022-11-28

**Authors:** Giuseppe Lassandro, Francesco Carriero, Domenico Noviello, Valentina Palladino, Giovanni Carlo Del Vecchio, Maria Felicia Faienza, Paola Giordano

**Affiliations:** 1Azienda Ospedaliero Universitaria Consorziale Policlinico-Giovanni XXIII, Pediatria Universitaria “B. Trambusti”, 70126 Bari, Italy; 2Interdisciplinay Medicine Department, Pediatric Section, University of Bari “Aldo Moro”, 70126 Bari, Italy; 3Department of Biomedical Science and Human Oncology, Pediatric Section, University of Bari “Aldo Moro”, 70126 Bari, Italy

**Keywords:** eltrombopag, children, MYH9

## Abstract

Inherited thrombocytopenias represents a heterogenous group of diseases characterized by a congenital reduction in the platelet count that could lead to a bleeding tendency. MYH9-related disorders are characterized by large platelets and congenital thrombocytopenia. Thrombopoietin-receptor agonists: eltrombopag and romiplostim are currently approved in many countries for the treatment of different forms of acquired thrombocytopenia, such as immune thrombocytopenia. We report, instead, the successful use of eltrombopag to treat inherited thrombocytopenia in a patient with an MHY9-related disease. This is the first report of a chronic use of eltrombopag to elevate platelets in MYH9-related disorders without side effects.

## 1. Introduction

Inherited thrombocytopenias (ITs) represent a heterogeneous group of diseases characterized by a congenital reduction in platelet number that could lead to a wide severity in bleeding symptoms [1]. ITs, once considered rare and obscure conditions, are today recognized with an increasing frequency thanks to improved knowledge. In fact, ITs have an estimated prevalence of 2.7 per 100,000 people in Italy [2]. Clinical spectrum ranges from asymptomatic to life-threatening bleeding. Moreover, patients with ITs sometimes need platelet transfusions before surgical interventions or invasive procedures because their platelet count is below the safe threshold for the specific procedures. ITs include non-syndromic and syndromic forms in which thrombocytopenia are associated with other clinical alterations. MYH9-related diseases (MYH9-RD) are characterized by enlarged platelets (more than 40% of >3.9 µm) and congenital thrombocytopenia (platelet count less than 150 × 10^9^/L). The most affected individuals develop one or more late-onset manifestations of the disorder, such as progressive sensorineural hearing loss, congenital cataract, an alteration of the liver enzymes and renal damage [3]. Thrombopoietin receptor agonists (TPO-RAs), eltrombopag and romiplostim, are currently approved for the treatment of acquired thrombocytopenias, such as immune thrombocytopenia (ITP). TPO-RAs, that mimic the action of thrombopoietin on its receptor and stimulate platelets and megakaryocytes’ production, are effective in reducing bleeding and avoiding the use of concomitant or rescue medications. Thrombopoietin receptor agonists may represent a promising treatment option for ITs, but to date there is little evidence on this topic [4]. TPO-RAs have demonstrated a favorable safety profile and high efficacy in pediatric ITP. However, different side effects as thrombotic events are reported. Additionally, they can be used as short-term treatments before surgical interventions or invasive procedures [5,6]. We report here the successful use of eltrombopag to treat thrombocytopenia in a child with an MHY9-related disease.

## 2. Case Presentation

A 10-year-old girl with a diagnosis of type 1 diabetes mellitus was referred to our department of paediatric haematology in May 2016. She presented with skin haemorrhagic manifestations and occasional findings of thrombocytopenia during the assessment for diabetes. At the presentation, our detailed physical examination revealed the presence of periorbital petechiae and bruising of the lower limbs. She was hospitalized and blood tests showed a severe thrombocytopenia (platelet count: 18 × 109/L, MPV: 9 fl) without significant morphological changes in the peripheral blood smear. The additional diagnostic tests, including renal, hepatic and thyroid functions, were normal, while most common infections were excluded. No alterations were detected in immunological tests except a transient positivity in antinuclear antibodies (homogeneous pattern, title 1/640) without additional clinical signs of systemic autoimmune disorder. A bone marrow examination was performed and showed “discrete cellularity, abundant megakaryocytes in different stages of maturation, some with slight dysmegakaryocytopoietic notes”. Suspecting immune thrombocytopenia, the patient was treated with intravenous immunoglobulins (IVIG) (0.8 g/kg) for 2 days, associated with methylprednisolone (20 mg/kg/day) for 3 days and then prednisone (1 mg/kg/day) with a partial response after 10 days of therapy (platelet count: 42 × 10^9^/L). Subsequently, during the hematologic follow-up the peripheral blood smear was repeated but did not reveal suggestive signs of giant platelet thrombocytopenia. She was periodically treated with IVIG and/or steroid therapy for platelet counts below 20 × 10/L. Given the partial response to these treatments, therapy with Eltrombopag (at minimal dose of 25 mg/day) was started. Surprisingly, after two weeks of therapy, the child’s platelet count, which had been persistently low, gradually increased. After 4 months, the patients’ platelet count increased to 134 × 10^9^/L without side effects (Figure 1). Otherwise, the partial response to IVIG induced clinicians to suspect an inherited form of thrombocytopenia. Sequencing analysis detected a heterozygous mutation c.2116C > G (p. Gln706Glu) in the MYH9 gene of the patient, but no mutations were found in the parents as a result of a de novo mutation [7].

## 3. Discussion

ITs are a heterogeneous group of congenital platelet disorders characterized by thrombocytopenia that could lead to bleeding and to syndromic and systemic manifestations [8]. Currently, ITs are poorly known by clinicians and often misdiagnosed with most common forms of thrombocytopenia. Our patient presented with hemorrhagic manifestations and severe thrombocytopenia. The presence of type 1 diabetes mellitus and the transient positivity of anti-nuclear antibodies suggested an initial diagnosis of ITP. Moreover, the peripheral blood smear did not reveal signs suggestive of giant platelet thrombocytopenia and the mean platelet volume (MPV) was at the upper limits. In ITs, diagnostic investigations are difficult and often requires a complex laboratory assessment available in only specialized centers. The peripheral blood smear appears useful in the diagnosis of ITs in ~25–30% of patients [9]. Moreover, some morphological features, such as an enlarged platelet, are non-specific and also described in acquired thrombocytopenia [10].

Additionally, the measurements of the platelet size present substantial difficulties. Noris et al. reported that the sensitivity and specificity of MPV in establishing platelet volume is widely variable depending on the instrument used. Furthermore, some of ITs may present platelets that, due to their increased size, are unrecognized by the electronic counter, which therefore underestimates the MPV [11]. In our case, the partial response to first-line ITP therapies raised the suspicion of an inherited genetic cause for thrombocytopenia. We performed a sequencing analysis that revealed a heterozygous mutation in the MYH9 gene. Despite the diagnosis of inherited thrombocytopenia, we decided to continue eltrombopag considering the successful clinical results reported in the literature for short time therapy. Unexpectedly, this drug stopped the patient’s bleeding symptoms and increased significantly the platelet count without adverse events. For these reasons, further studies could encourage the use of eltrombopag in the treatment of different types of thrombocytopenia. Eltrombopag is a small-molecule orally bioavailable that binds a transmembrane site of the TPO receptor and activates a cascade of signaling events that finally stimulates megakaryocytopoiesis and platelet production. Recently, eltrombopag has been approved for the treatment of acquired thrombocytopenia. However, studies regarding its safety and efficacy in treating inherited thrombocytopenia are still ongoing. A recent review considered several clinical studies on the use of TPO-RAs in ITs. The authors believe that TPO-RAs should be used for the treatment of different forms of ITs after carefully evaluating the risk/benefit for each patient [12]. Zaninetti et al. conducted a phase 2 clinical trial that evaluates the short- and long-term eltrombopag use in preparing patients with ITs for invasive procedures or in reducing significant bleeding symptoms. This study illustrates that an eltrombopag treatment induces an increase in platelet counts in several forms of ITs, especially in the patient with MYH9-RD [4]. In another trial, twelve patients with MYH9-RD and thrombocytopenia (platelet count < 50 × 10^9^/L) received eltrombopag for 3–6 weeks. The treatment was well tolerated and the bleeding symptoms disappeared in most patients [13]. However, the role of eltrombopag in ITs is reported only for short periods to allow for the rising of the platelet count before surgery [14,15]. This is the first experience of the long-term use of eltrombopag. Based on their mechanism of action and widespread application in acquired thrombocytopenias, TPO-RAs may represent one highly encouraging, non-invasive approach to raising the platelet count. Several clinical experiences, including our case report, have shown that TPO-RAs are well tolerated and effective, but further research is required to provide evidence of the effectiveness and feasibility of TPO-RAs in patients affected by ITs as well as to optimize treatment dose and timing.

## Figures and Tables

**Figure 1 children-09-01839-f001:**
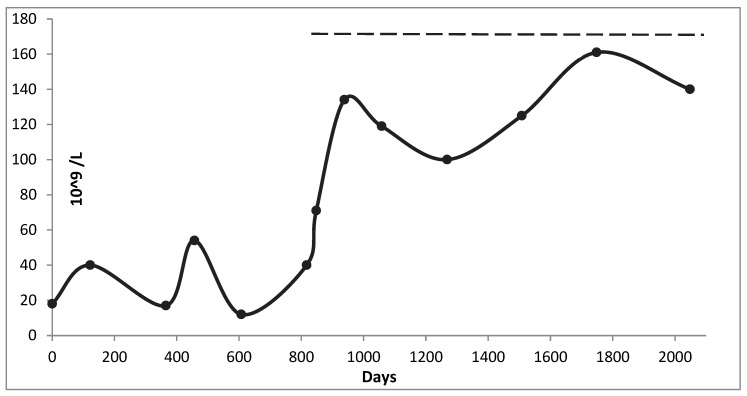
Platelet count during follow-up.

## Data Availability

The data are kept by the author and can be requested by email.

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
