# Peer review of "Successful Eltrombopag Therapy in a Child with MYH9-Related Inherited Thrombocytopenia"

_children, 2022, doi:10.3390/children9121839_

Round 1

Reviewer 1 Report

This paper describes a new pediatric case of successful MYH9 related thrombocytopenia therapy with elthrombopag. although not completely new, this case report highlights the potential benefits of eltrombopag in patients, even pediatric ones, when suffering severe inherited thrombocytopenia

I have some suggestions and more stringent requests regarding the paper:

- I suggest to mention in the discussion the first pediatric published case by R Favier in 2013  https://doi.org/10.1542/peds.2012-3807 

It is not clear from the case report and from the figure if the initial posology of 25mg/d has been later lowered. Given the uncertainties about long term use of elthrombopag in children, the authors should discuss the option to further diminish to a minimal dose allowing the platelets to stay above 50G/l; this can be achieved for instance by giving 25mg elthrombopag every other day.

major request: there are several citations which have no relevance to the subject and thus should be removed, namely ref 6,10,11,12

there are several english misspelling to be corrected, ie on line 64,68,71,79,84, 86,93,108

Author Response

A: I suggest to mention in the discussion the first pediatric published case by R Favier in 2013  https://doi.org/10.1542/peds.2012-3807 

R: Done

A: It is not clear from the case report and from the figure if the initial posology of 25mg/d has been later lowered. Given the uncertainties about long term use of elthrombopag in children, the authors should discuss the option to further diminish to a minimal dose allowing the platelets to stay above 50G/l; this can be achieved for instance by giving 25mg elthrombopag every other day.

R:  We have included in the text that we started at the minimum recognized dose of 25 mg. We prefer not to insert anything else because the official therapeutic indications for ITP do not provide for a reduction in the dose or frequency of administration

A: major request: there are several citations which have no relevance to the subject and thus should be removed, namely ref 6,10,11,12

R: We remove 11 and 12, 6 is important for the management of thrombosis in children and 1o is important to underline the demanding hematological patients

A: there are several english misspelling to be corrected, ie on line 64,68,71,79,84, 86,93,108

R: done

Reviewer 2 Report

The manuscript by Lassandro et al, describes a case report of a patient with MYH9-RD and diabetes, and how treatment with Eltrombopag, a TPO-receptor agonist, improved the platelet count, and reduced the bleeding diatheses presented at baseline.

The manuscript is interesting, but requires extensive English editing. Many ideas and concepts are just mentioned, but the authors need to connect those ideas or descriptions, and contextualize their conclusions. 

Some questions and concerns:

Was there not a blood smear prepared after the low platelet count was detected with the hemocytocounter? What was the MPV value? That would have given information against the initial ITP diagnosis. Please discuss about this in the Discussion. It is very important, as many patients receive unnecessary treatments.

Figure 1: Show also the MPV and PDW in parallel to the platelet counts. In ITP, the MPV should be restored after treatment, in responders, but in a congenital pathology such as MYH9-RD, the MPV should remain LARGE, with potential anysocitosis (wide PDW). If the hemocytocounter reads the IPF, show as well. It might be an interesting variable in this dynamic overview.

Mutation: Please describe that the mutation has been already reported before, see ref: 10.1080/09537104.2017.1294250 

As mentioned before, blood smear observation is essential in these patients. Show pics, if you have them, otherwise, discuss that it should have been done. 

Please discuss the fact that the bleeding in this patient may also be enhanced or conditioned by the accompanying diabetes. In this regard, was there a functional test performed after platelet count recovery? Please comment on this, or add if it was performed.

Extend in the Discussion that TPO-RAs are only approved for the treatment of chronic ITP in adults, and few other clinical scenarios with thrombocytopenia... and that evidence is required for the extension of their indication to other pathologies that can benefit from them. It is more or less there, but the contextualization lacks.

The following publication can help, and authors should check the current updates:

The use of TPO-RAs to other applicacionts than ITP, some cases of aplastic anemia, or throbocytopenia associated to hepatitis... in adults. Only Romiplostim has approved indication in children with ITP. TPO-RAs are a promising therapeutic option that requires evidence (https://www.ncbi.nlm.nih.gov/pmc/articles/PMC8318934/).

-------

Other details:

Line 34, ... x 109/L..., 9 should be superscript.

Line 42, "there are few evidence".. I would suggest "ther is little evidence"...

Lines 42 to the end of the Intro: please write connecting ideas, it reads like a copy-paste of concepts that were not properly connected.

Line 66, the platelet unit is missing the "to the 9", 10E9/L.

The last sentence of the manuscript is not clear. Please contextualize.

Author Response

A: Was there not a blood smear prepared after the low platelet count was detected with the hemocytocounter? What was the MPV value? That would have given information against the initial ITP diagnosis. Please discuss about this in the Discussion. It is very important, as many patients receive unnecessary treatments.

R: in the introduction we added that the peripheral smear was performed and there were no suggestive volumetric anomalies (confirmed by MPV at hemocytocounter). In the discussion we undeline the importance to repeat smear to avoid misdiagnosis

A: Figure 1: Show also the MPV and PDW in parallel to the platelet counts. In ITP, the MPV should be restored after treatment, in responders, but in a congenital pathology such as MYH9-RD, the MPV should remain LARGE, with potential anysocitosis (wide PDW). If the hemocytocounter reads the IPF, show as well. It might be an interesting variable in this dynamic overview.

R: MPV was normal and is difficult to show it 

A: Mutation: Please describe that the mutation has been already reported before, see ref: 10.1080/09537104.2017.1294250 

R: done

A: As mentioned before, blood smear observation is essential in these patients. Show pics, if you have them, otherwise, discuss that it should have been done. 

R: no pics available 

A: Please discuss the fact that the bleeding in this patient may also be enhanced or conditioned by the accompanying diabetes. In this regard, was there a functional test performed after platelet count recovery? Please comment on this, or add if it was performed.

R: No functional test performed. Diabetes in this case do not worse the bleeding

A: Extend in the Discussion that TPO-RAs are only approved for the treatment of chronic ITP in adults, and few other clinical scenarios with thrombocytopenia... and that evidence is required for the extension of their indication to other pathologies that can benefit from them. It is more or less there, but the contextualization lacks.

R: TPO-RAs are approved also for children with ITP. In this case patient start Eltrombopag for a misdiagnosis.

A: The following publication can help, and authors should check the current updates: The use of TPO-RAs to other applicacionts than ITP, some cases of aplastic anemia, or throbocytopenia associated to hepatitis... in adults. Only Romiplostim has approved indication in children with ITP. TPO-RAs are a promising therapeutic option that requires evidence (https://www.ncbi.nlm.nih.gov/pmc/articles/PMC8318934/).

R: see above. Eltrombopag is just approved for children with ITP (as romiplostin)

Round 2

Reviewer 1 Report

now suitable for publication

please correct line 76:  as cause of ...

Author Response

We correct line 76, we improve english, we enrich the text
